# Uncertainty Estimation Using Riemannian Model Dynamics for Offline Reinforcement Learning

**Guy Tennenholtz**[*]
Technion Institute of Technology

**Shie Mannor**
Technion Institute of Technology & Nvidia Research

## Abstract

Model-based offline reinforcement learning approaches generally rely on bounds of model error. Estimating these bounds is usually achieved through uncertainty estimation methods. In this work, we combine parametric and nonparametric methods for uncertainty estimation through a novel latent space based metric. In particular, we build upon recent advances in Riemannian geometry of generative models to construct a pullback metric of an encoder-decoder based forward model. Our proposed metric measures both the quality of out-of-distribution samples as well as the discrepancy of examples in the data. We leverage our method for uncertainty estimation in a pessimistic model-based framework, showing a significant improvement upon contemporary model-based offline approaches on continuous control and autonomous driving benchmarks.

## 1 Introduction

Offline Reinforcement Learning (RL) [Levine et al., 2020], a.k.a. batch-mode RL [Ernst et al., 2005, Riedmiller, 2005, Fonteneau et al., 2013], involves learning a policy from data sampled by a potentially suboptimal policy. Offline RL seeks to *surpass* the average performance of the agents that generated the data. Traditional methodologies fall short in offline settings, causing overestimation of the return [Buckman et al., 2020, Wang et al., 2020, Zanette, 2020].

One approach to overcome this in model-based settings is to penalize the return in out of distribution (OOD) regions, as depicted in Figure 1. In this manner, the agent is constrained to stay "near" areas of low model error, thereby limiting possible overestimation. However, reliable estimates of model error are key to the success of such methods.

Estimating model error in OOD regions can be achieved through uncertainty estimation [Yu et al., 2020]. Methods of parametric uncertainty estimation such as bootstrap ensembles [Efron, 1982], Monte Carlo Dropout [Gal and Ghahramani, 2016], and randomized priors [Osband et al., 2018], may be susceptible to poor model specification and are most effective when dealing with large datasets. In contrast, nonparametric methods such as k-nearest neighbors (k-NN) [Villa Medina et al., 2013, Fathabadi et al., 2021] are beneficial in regions of limited data, yet require a proper metric to be used.

We propose to combine parametric and nonparametric methods for uncertainty estimation. Particularly, we define a novel Riemmannian metric which captures the epistemic and aleatoric uncertainty of a generative parametric forward model. This distance metric is then applied to measure the average geodesic distance to the $k$-nearest neighbors in the data. We derive analytical expressions for our metric and provide an efficient manner to estimate it. We then demonstrate the effectiveness of our metric for penalizing an offline RL agent compared to contemporary approaches on continuous control and autonomous driving benchmarks. As we empirically show, common approaches, including statistical bootstrap ensemble or Euclidean distances in latent space, do not necessarily capture the underlying degree of error needed for model-based offline RL.

---

[*]Correspondence to guytenn@gmail.com

36th Conference on Neural Information Processing Systems (NeurIPS 2022).

## 2  Preliminaries

### 2.1  Offline Reinforcement Learning

We consider the standard Markov Decision Process (MDP) framework [Puterman, 2014] defined by the tuple $(\mathcal{S}, \mathcal{A}, r, P, \alpha)$, where $\mathcal{S}$ is the state space, $\mathcal{A}$ the action space, $r : \mathcal{S} \times \mathcal{A} \mapsto [0, 1]$ the reward function, $P : \mathcal{S} \times \mathcal{A} \times \mathcal{S} \mapsto [0, 1]$ the transition kernel, and $\alpha \in (0, 1)$ is the discount factor.

In the online setting of reinforcement learning (RL), the environment initiates at some state $s_0 \sim \rho_0$. At any time step the environment is in a state $s \in \mathcal{S}$, an agent takes an action $a \in \mathcal{A}$ and receives a reward $r(s, a)$ from the environment as a result of this action. The environment transitions to state $s'$ according to the transition function $P(\cdot|s, a)$. The goal of online RL is to find a policy $\pi(a|s)$ that maximizes the expected discounted return $v^\pi = \mathbb{E}_\pi \left[ \sum_{t=0}^{\infty} \alpha^t r(s_t, a_t)|s_0 \sim \rho_0 \right]$.

Unlike the online setting, the offline setup considers a dataset $\mathcal{D}_n = \{s_i, a_i, r_i, s'_i\}_{i=1}^n$ of transitions generated by some unknown agents. The objective of offline RL is to find the best policy in the test environment (i.e., real MDP) given only access to the data generated by the unknown agents.

### 2.2  Riemannian Manifolds

We define the Riemannian pullback metric, a fundamental component of our proposed method. We refer the reader to Carmo [1992] for further details on Riemannian geometry.

We are interested in studying a smooth surface $M$ with a Riemannian metric $g$. A Riemannian metric is a smooth function that assigns a symmetric positive definite matrix to any point in $M$. At each point $z \in M$ a tangent space $T_z M$ specifies the pointing direction of vectors "along" the surface.

**Definition 1.** *Let $M$ be a smooth manifold. A Riemannian metric $g$ on $M$ changes smoothly and defines a real scalar product on the tangent space $T_z M$ for any $z \in M$ as*

$$g_z(x, y) = \langle x, y \rangle_z = \langle x, G(z)y \rangle, \quad x, y \in T_z M,$$

*where $G(z) \in \mathbb{R}^{d_z \times d_z}$ is the corresponding metric tensor. $(M, g)$ is called a Riemannian manifold.*

The Riemannian metric enables us to easily define geodesic curves. Consider some differentiable mapping $\gamma : [0, 1] \mapsto M \subseteq \mathbb{R}^{d_z}$, such that $\gamma(0) = z_0, \gamma(1) = z_1$. The length of the curve $\gamma$ measured on $M$ is given by

$$L(\gamma) = \int_0^1 \sqrt{\left\langle \frac{\partial \gamma(t)}{\partial t}, G(\gamma(t)) \frac{\partial \gamma(t)}{\partial t} \right\rangle} dt. \quad (1)$$

The geodesic distance $d(z_1, z_2)$ between any two points $z_1, z_2 \in M$ is then the infimum length over all curves $\gamma$ for which $\gamma(0) = z_0, \gamma(1) = z_1$. That is, $d(z_1, z_2) = \inf_\gamma L(\gamma)$ s.t. $\gamma(0) = z_0, \gamma(1) = z_1$. The geodesic distance can be found by solving a system of nonlinear ordinary differential equations (ODEs) defined in the intrinsic coordinates [Carmo, 1992].

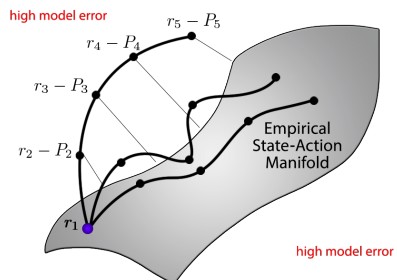

Figure 1: Illustration of a data manifold corresponding to model error. Curved lines represent trajectories at different stages of training. Agent incurs a penalty in areas of high model error.

**Pullback Metric.** Assume an ambient (observation) space $\mathcal{X}$ and its respective Riemannian manifold $(M_{\mathcal{X}}, g_{\mathcal{X}})$. Learning $g_{\mathcal{X}}$ can be hard (e.g., learning the distance metric between images). Still, it may be captured through a low dimensional submanifold. As such, it is many times convenient to parameterize the surface $M_{\mathcal{X}}$ by a latent space $\mathcal{Z} = \mathbb{R}^{d_z}$ and a smooth function $f : \mathcal{Z} \mapsto \mathcal{X}$, where $\mathcal{Z}$ is a low dimensional latent embedding space. As learning the manifold $M_{\mathcal{X}}$ can be hard, we turn to learning the immersed low dimensional submanifold $M_{\mathcal{Z}}$ (for which the chart maps are trivial, since $\mathcal{Z} = \mathbb{R}^{d_z}$). Given a curve $\gamma : [0, 1] \mapsto M_{\mathcal{Z}}$ we have that $\left\langle \frac{\partial f(\gamma(t))}{\partial t}, G_{\mathcal{X}}(f(\gamma(t))) \frac{\partial f(\gamma(t))}{\partial t} \right\rangle = \left\langle \frac{\partial \gamma(t)}{\partial t}, J_f^T(\gamma(t)) G_{\mathcal{X}}(f(\gamma(t))) J_f(\gamma(t)) \frac{\partial \gamma(t)}{\partial t} \right\rangle$, where the Jacobian matrix $J_f(z) = \frac{\partial f}{\partial z} \in \mathbb{R}^{d_{\mathcal{X}} \times d_z}$

maps tangent vectors in $TM_{\mathcal{Z}}$ to tangent vectors in $TM_{\mathcal{X}}$. The induced metric is thus given by

$$G_f(z) = J_f(z)^T G_{\mathcal{X}}(f(z)) J_f(z). \tag{2}$$

The metric $G_f$ is known as the *pullback metric*, as it "pulls back" the metric $G_{\mathcal{X}}$ on $\mathcal{X}$ back to $G_f$ via $f : \mathcal{Z} \mapsto \mathcal{X}$. The pullback metric captures the intrinsic geometry of the immersed submanifold while taking into account the ambient space $\mathcal{X}$. The geodesic distance in ambient space is captured by geodesics in the latent space $\mathcal{Z}$, reducing the problem to learning the latent embedding space $\mathcal{Z}$ and the observation function $f$. Indeed, learning the latent space and observation function $f$ can be achieved through a encoder-decoder framework, such as a VAE [Arvanitidis et al., 2018].

## 3   Background: Penalty of Uncertainty for Offline Reinforcement Learning

A key element of model-based RL methods involves estimating a model $\hat{P}(s'|s,a)$ to construct a pessimistic MDP[2]. This work builds upon MOPO, a recently proposed model-based offline RL framework [Yu et al., 2020]). Particularly, we assume access to an approximate MDP $(\mathcal{S}, \mathcal{A}, \hat{r}, \hat{P}, \alpha)$ (e.g., trained by maximizing the likelihood of the data), and define a penalized MDP $(\mathcal{S}, \mathcal{A}, \tilde{r}, \hat{P}, \alpha)$, such that for all $s \in \mathcal{S}, a \in \mathcal{A}$, $\tilde{r}(s,a) = \hat{r}(s,a) - \lambda c(P(\cdot|s,a), \hat{P}(\cdot|s,a))$, where $c$ penalizes the reward according to model error (e.g., the total variation distance) and $\lambda > 0$. The offline RL problem is then solved by executing an online algorithm in the reward-penalized MDP. Unfortunately, as $P(\cdot|s,a)$ is unknown, and can only be estimated from the data, $c(P(\cdot|s,a), \hat{P}(\cdot|s,a))$ cannot be calculated. Nevertheless, one can attempt to upper bound the distance, i.e., for some $U : \mathcal{S} \times \mathcal{A} \mapsto \mathbb{R}$, $c(P(\cdot|s,a), \hat{P}(\cdot|s,a)) \leq U(s,a), \forall s \in \mathcal{S}, a \in \mathcal{A}$. In this work we propose to use a naturally induced metric of a variational forward model, which we show can introduce an effective penalty for offline RL. In Section 4 we define this metric, and finally, we leverage it in Section 4.4.

## 4   Metrics of Uncertainty

As described in the previous section, our goal is to estimate model error in order to penalize the agent in out of distribution (OOD) regions. Yu et al. [2020] proposed to achieve this through bootstrap ensembles, an out of distribution uncertainty estimation technique. Alternatively, we propose to employ a well-known nonparametric approach for uncertainty estimation [Villa Medina et al., 2013, Fathabadi et al., 2021], namely $k$-nearest neighbors ($k$-NN). Specifically, for any $s, a \in \mathcal{S} \times \mathcal{A}$, we estimate model error by

$$U(s,a) = \frac{1}{k} \sum_{(s_i, a_i) \in \text{NN}_k(s,a)} d((s,a), (s_i, a_i)), \tag{3}$$

where $d : \mathcal{S} \times \mathcal{A} \times \mathcal{S} \times \mathcal{A} \mapsto \mathbb{R}_+$ is a distance metric, and $\text{NN}_k(s,a)$ is the set of $k$-nearest neighbors of $(s,a)$ in $\mathcal{D}_n$ according to the distance metric $d$.

A question arises: how to choose $d$? Using the Euclidean distance in ambient (state-action) space is usually a bad choice (e.g., the $\ell_2$ distance between natural images is not necessarily meaningful). Moreover, to correctly measure the error, model dynamics should be somehow taken into consideration. We therefore consider an alternative approach which leverages the latent space of a variational forward model, as described next.

### 4.1   A Variational Latent Model of Dynamics

We begin by modeling $\hat{P}(s'|s,a)$ using a generative latent model. Specifically, we consider a latent model which consists of an encoder $E : \mathcal{S} \times \mathcal{A} \mapsto \mathcal{B}(\mathcal{Z})$ and a decoder $f_D : \mathcal{Z} \mapsto \mathcal{B}(\mathcal{S})$, where $\mathcal{B}(\mathcal{X})$ is set of probability measures on the Borel sets of $\mathcal{X}$. While the encoder $E$ learns a latent representation of $s, a$, the decoder $f_D$ estimates the next state $s'$ according to $P(\cdot|s,a)$. This model corresponds to the decomposition $P(\cdot|s,a) = f_D(\cdot|E(s,a))$. Such a model can be trained by maximizing the Evidence Lower BOund (ELBO, Kingma and Welling [2013]) over the data.

---

[2]Pessimism is a key element of offline RL algorithms [Jin et al., 2020], limiting overestimation of a trained policy due to the distribution shift between the data and the trained policy.

---

**Algorithm 1** GELATO: Geometrically Enriched LATent model for Offline reinforcement learning

---
1: **Input:** Offline dataset $\mathcal{D}_n$, RL algorithm
2: Train variational latent forward model on dataset $\mathcal{D}_n$ by maximizing ELBO.
3: Construct approximate MDP $(\mathcal{S}, \mathcal{A}, \hat{r}, \hat{P}, \alpha)$
4: Use distance $d_{\mathcal{Z}}$ induced by pullback metric $G_{f_D, U \circ f_F}$ (Theorem 2) to penalize reward $\tilde{r}_d(s, a) = \hat{r}(s, a) - \lambda U(s, a)$ where $U(s, a) = \sum_{(s_i, a_i) \in \text{NN}_k(s, a)} d_{\mathcal{Z}}(E(s, a), E(s_i, a_i))$
5: Train RL algorithm over penalized MDP $(\mathcal{S}, \mathcal{A}, \tilde{r}_d, \hat{P}, \alpha)$

---

That is, given a prior $P(z)$, we model $E_\phi, f_{D,\theta}$ as parametric functions and maximize the ELBO, $\max_{\theta, \phi} \mathbb{E}_{E_\phi(z|s,a)} \left[ \log f_{D,\theta}(s'|z) \right] - D_{KL}(E_\phi(z|s,a)||P(z))$ We refer the reader to the appendix for an exhaustive overview of training VAEs by maximum likelihood and the ELBO.

Recall that we wish to find a good metric for estimating model error. Having learned a latent model for $\hat{P}(s'|s, a)$, its latent space $\mathcal{Z}$ can be used to define the metric $d$ in Equation (3), i.e., the Euclidean distance between latent representations of state-action pairs. Unfortunately, as was previously shown [Arvanitidis et al., 2018], latent codes in variational models contain sharp discontinuities, rendering Euclidean distances in latent space unreliable and inaccurate (as we will also demonstrate in our experiments). Instead, we propose to use the natural induced metric of our latent model, as described in the following subsection.

### 4.2 The Pullback Metric of Model Dynamics

In this part we define the metric $d$ that will be used to approximate model error in Equation (3). Specifically, we consider a Riemannian submanifold defined by a latent space $\mathcal{Z}$ and observation function $f$, which induces minimum energy in the ambient space. We will later choose $\mathcal{Z}$ to be the latent space of our variational model (i.e., encoded state-action) and $f$ to be the decoder function $f_D$ of next state transitions. We define the distance metric formally below.

**Definition 2.** *We define a Riemannian submanifold $(\mathcal{M}_{\mathcal{Z}}, g_{\mathcal{Z}})$ by a differential function $f : \mathcal{Z} \mapsto \mathcal{S}$ and latent space $\mathcal{Z}$ such that*

$$d_{\mathcal{Z}}(z_1, z_2) = \inf_\gamma \int_0^1 \left\| \frac{\partial f(\gamma(t))}{\partial t} \right\| dt \quad s.t. \ \gamma(0) = z_1, \gamma(1) = z_2.$$

A similar metric has been used in previous work on generative latent models [Chen et al., 2018, Arvanitidis et al., 2018]. By choosing $f$ to be the decoder function $f_D$, latent codes that are close w.r.t. $d_{\mathcal{Z}}$ induce curves of minimal energy in the ambient observation space (i.e., next state). This metric is closely related to the pullback metric (see Section 2.2), as shown by the following proposition.

**Proposition 1.** *Let $(\mathcal{M}_{\mathcal{Z}}, g_{\mathcal{Z}})$ as defined above. Then $G_f(z) = J_f^T(z) J_f(z)$, for any $z \in \mathcal{Z}$.*

Indeed, Proposition 1 shows us that $G_f$ is a pullback metric. Particularly $\mathcal{Z}$ and $J_f$ define the structure of the ambient observation space $\mathcal{X}$ (in our case, next state transitions).

By choosing $f$ to be the decoder function $f_D$, the metric $G_{f_D}$ becomes stochastic, complicating analysis. Instead, as proposed and analyzed in Arvanitidis et al. [2018], we use the expected pullback metric $\mathbb{E}[G_{\mathcal{Z}}]$ as an approximation of the underlying stochastic metric. Similar to previous work on variational models, we use a normally distributed decoder to define the output. Using Proposition 1, we have the following result (see Appendix for proof).

**Theorem 1.** *[Arvanitidis et al. [2018]] Assume $f_D(\cdot|z) \sim \mathcal{N}(\mu(z), \sigma(z)I)$. Then*

$$\mathbb{E}_{f_D(\cdot|z)} \left[ G_{f_D}(z) \right] = G_\mu(z) + G_\sigma(z), \tag{4}$$

*where $G_\mu(z) = J_\mu^T(z) J_\mu(z)$ and $G_\sigma(z) = J_\sigma^T(z) J_\sigma(z)$.*

Given an embedded latent space $\mathcal{Z}$, the expected metric in Equation (4) gives us a sense of the topology of the latent space manifold induced by $f_D$. The terms $G_\mu = J_\mu^T J_\mu$ and $G_\sigma = J_\sigma^T J_\sigma$ are in fact the induced pullback metrics of $\mu$ and $\sigma$, respectively.

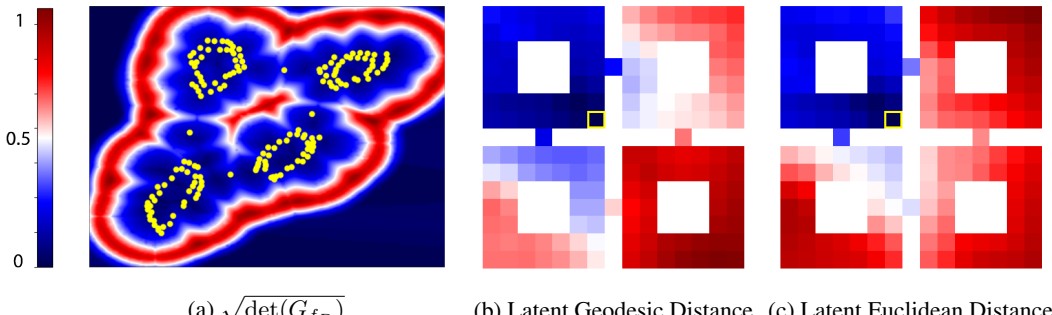

(a) $\sqrt{\det(G_{f_D})}$      (b) Latent Geodesic Distance    (c) Latent Euclidean Distance

Figure 2: **(a)** The latent space (yellow markers) of the grid world environment and the geometric volume measure of the decoder-induced metric (background). **(b, c)** The geodesic distance of the decoder-induced submanifold and the Euclidean distance of latent states, as viewed in ambient space. All distances are calculated w.r.t. the yellow marked state. **Note**: colors in (a), which measure magnitude, are unrelated to colors in (b,c), which measure distance to the yellow marked state.

## 4.3 Capturing Epistemic and Aleatoric Uncertainty

The previously proposed encoder-decoder model induces a metric which captures the structure of the learned dynamics. However, the decoder variance, $\sigma(z)$, does not differentiate between aleatoric uncertainty (environment dynamics) and epistemic uncertainty (missing data).

We propose two methods to enrich the metric in Equation (4) in order to achieve a better estimate of uncertainty. First, by using an ensemble of $M$ decoder functions $\{f_{D,i}\}_{i=1}^{M}$ trained using standard bootstrap techniques [Efron, 1982], we capture the traditional epistemic uncertainty of the decoder parameters. Second, to correctly distinguish epistemic and aleatoric uncertainty, we add a latent forward function to our previously proposed variational model. Specifically, our latent model consists of an encoder $E : \mathcal{S} \times \mathcal{A} \mapsto \mathcal{B}(\mathcal{Z})$, forward model $f_F : \mathcal{Z} \mapsto \mathcal{B}(\mathcal{X})$ and decoder functions $f_{D,i} : \mathcal{X} \mapsto \mathcal{B}(\mathcal{S})$ such that $P(\cdot|s,a) = f_{D,i}(\cdot|x)$, and $x \sim f_F(\cdot|E(s,a))$. This structure enables us to capture the aleatoric uncertainty under the forward transition model $f_F$, and the epistemic uncertainty using the decoders $f_{D,i}$. That is, for $f_F(\cdot|z) \sim \mathcal{N}(\mu_F(z), \sigma_F(z)I)$, the variance model $\sigma_F(z)$ captures the stochasticity in model dynamics. This decomposition is also helpful whenever one wants to train an agent in latent space (e.g., for planning Schrittwieser et al. [2020])

Next, we turn to analyze the pullback metric induced by the proposed forward transition model. As both $f_F$ and $\{f_{D,i}\}_{i=1}^{m}$ are stochastic (capturing epistemic and aleatoric uncertainty), the result of Theorem 1 cannot be directly applied to their composition. The following proposition provides an analytical expression for the expected pullback metric of a sampled next state and a uniformly sampled decoder (the proof is given in the appendix).

**Theorem 2.** *Assume* $f_F(\cdot|z) \sim \mathcal{N}(\mu_F(z), \sigma_F(z)I)$, $f_{D,i}(\cdot|x) \sim \mathcal{N}(\mu_D^i(x), \sigma_D^i(x)I)$, $U \sim Unif\{1, \dots, M\}$. *Then, the expected pullback metric of the composite function* $(f_{D,U} \circ f_F)$ *is given by*

$$\mathbb{E}_{P(f_{D,U} \circ f_F)}\Big[ G_{f_{D,U} \circ f_F}(z) \Big] = J_{\mu_F}^T(z) \overline{G}_{f_D}(z) J_{\mu_F}(z) + J_{\sigma_F}^T(z) diag\big(\overline{G}_{f_D}(z)\big) J_{\sigma_F}(z),$$

*where* $\overline{G}_{f_D}(z) = \frac{1}{M} \sum_{i=1}^{M} \mathbb{E}_{x \sim F(\cdot|z)}\Big[ J_{\mu_D^i}^T(x) J_{\mu_D^i}(x) + J_{\sigma_D^i}^T(x) J_{\sigma_D^i}(x) \Big]$.

Unlike the metric in Equation (4), the composite metric distorts the decoder metric with Jacobian matrices of the forward model statistics. It takes into account both the aleatoric and epistemic uncertainty through the forward model as well as ensemble of decoders. As a special case we note the metric for the case of deterministic model dynamics.

**Corollary 1.** *Assume deterministic model dynamics, i.e.,* $x = f_F(z)$, *and without loss of generality assume* $f_F \equiv I$. *Then, the expected pullback metric of Theorem 2 is given by* $\mathbb{E}\Big[ G_{f_{D,U} \circ f_F}(z) \Big] = \frac{1}{M} \sum_{i=1}^{M} J_{\mu_D^i}^T(z) J_{\mu_D^i}(z) + J_{\sigma_D^i}^T(z) J_{\sigma_D^i}(z)$.

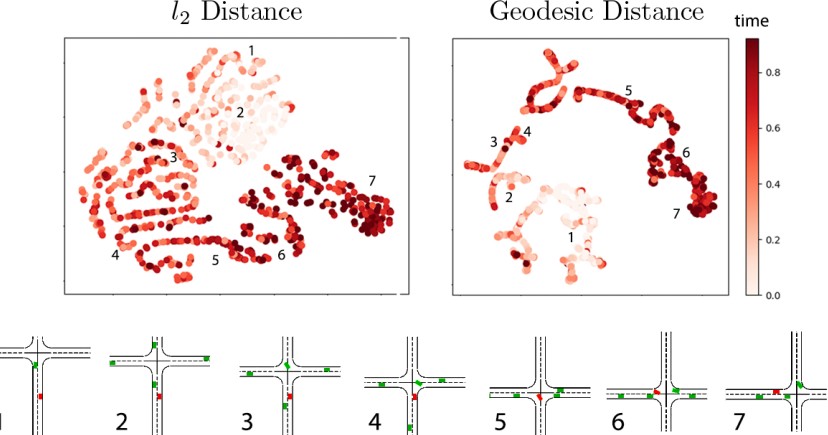

Figure 3: Plots show t-SNE embeddings generated in the intersection environment. Left plot depicts embeddings using Euclidean distances. Right plot depicts embeddings using geodesics distances which induce curves of minimum energy in ambient space. Colors correspond to the time in a trajectory (normalized w.r.t. longest trajectory). Numbering show mappings of a specific trajectory's states onto the embedded space as the controlled red car takes a left turn at an intersection (trajectory visualization shown under plots).

### 4.4 GELATO: Geometrically Enriched LATent model for Offline reinforcement learning

Having defined our metric, we are now ready to leverage it in a model based offline RL framework. Specifically, provided a dataset $\mathcal{D}_n = \{(s_i, a_i, r_i, s'_i)\}_{i=1}^n$ we train the variational latent forward model described in the previous section.

Algorithm 1 presents GELATO, our proposed approach. In GELATO, we estimate model error by measuring the distance of a new sample to the data manifold. We construct the reward-penalized MDP for which the error acts as a pessimistic regularizer. Finally, we train an RL agent over the pessimistic MDP with transition $\hat{P}(\cdot|s, a)$ and reward $r(s, a) - \lambda U(s, a)$. By achieving an improved estimate for model error the model-based pessimistic approach can significantly improve performance, as shown in the following section.

## 5 Experiments

This section is dedicated to quantitatively and qualitatively understand the benefits of our proposed metric and method. We validate two principal claims: **(1) Our metric captures inherent characteristics of model dynamics.** We demonstrate this by visualizing state-action geodesics of a toy grid world problem and a multi-agent autonomous driving task. We show that curves of minimum energy in ambient space indeed capture intrinsic properties of the problem. **(2) Our metric provides an improved OOD uncertainty estimate for offline RL.** We compare the traditionally used bootstrap ensemble method to our approach, which leverages our pullback metric in a nonparametric nearest neighbors approach. We also compare our method to simple use of Euclidean distances in latent space. We run extensive experiments on continuous control and autonomous driving benchmarks. We show that our metric achieves significantly improved performance in tasks for which geodesics are non-euclidean.

### 5.1 Metric Visualization

**Four Rooms.** To better understand the inherent structure of our metric, we constructed a grid-world environment for visualizing our proposed latent representation and metric. The $15 \times 15$ environment, as depicted in Figure 2, consists of four rooms, with impassable obstacles in their centers. The agent, residing at some position $(x, y) \in [-1, 1]^2$ in the environment can take one of four actions: up, down, left, or right – moving the agent $1, 2$ or $3$ steps (uniformly distributed) in that direction. We collected a dataset of 10000 samples, taking random actions at random initializations of the environment. The ambient state space was represented by the position of the agent – a vector of dimension 2, normalized to values in $[-1, 1]$. Finally, we trained a variational latent model with latent dimension $d_{\mathcal{Z}} = 2$.

Table 1: Performance of GELATO and its variants in comparison to contemporary model-based methods on D4RL datasets. Scores correspond to the return, averaged over 5 seeds (standard deviation removed due to space constraints and is given in the appendix). Results for MOPO, MBPO, SAC, and imitation are taken from Yu et al. [2020]. Mean score of dataset added for reference. Bold scores show an improved score w.r.t other methods.

| | Hopper | | | Walker2d | | | Halfcheetah | | |
|---|---|---|---|---|---|---|---|---|---|
| Method | Rand | Med | Med-Expert | Rand | Med | Med-Expert | Rand | Med | Med-Expert |
| Data Score | 299 | 1021 | 1849 | 1 | 498 | 1062 | -303 | 3945 | 8059 |
| GELATO (metric) | **685** | **1676** | 574 | **412** | **1269** | **1515** | 2560 | **5168** | **7989** |
| GELATO ($\ell_2$) | 544 | 1320 | 815 | 388 | 312 | 1255 | 512 | 4096 | 7304 |
| MOPO (bootstrap) | **677** | 1202 | 1063 | **396** | 518 | 1296 | **4114** | 4974 | 7594 |
| MBPO | 444 | 457 | 2105 | 251 | 370 | 222 | 3527 | 3228 | 907 |
| SAC | 664 | 325 | 1850 | 120 | 27 | -2 | 3502 | -839 | -78 |
| Imitation | 615 | 1234 | **3907** | 47 | 193 | 329 | -41 | 4201 | 4164 |

We used a standard encoder $z \sim \mathcal{N}(\mu_\theta(s), \sigma_\theta(s))$ and decoder $s' \sim \mathcal{N}(\mu_\phi(z), \sigma_\phi(z))$ represented by neural networks trained end-to-end using the evidence lower bound. We refer the reader to the appendix for an exhaustive description of the training procedure.

The latent space output of our model is depicted by yellow markers in Figure 2a. Indeed, the latent embedding consists of four distinctive clusters, structured in a similar manner as our grid-world environment. Interestingly, the distortion of the latent space accurately depicts an intuitive notion of distance between states. As such, rooms are distinctively separated, with fair distance between each cluster. States of pathways between rooms clearly separate the room clusters, forming a topology with four discernible bottlenecks.

In addition to the latent embedding, Figure 2a depicts the geometric volume measure $\sqrt{\det(G_{f_D})}$ of the trained pullback metric induced by $f_D$. This quantity demonstrates the effective geodesic distances between states in the decoder-induced submanifold. Indeed geodesics between data points to points outside of the data manifold (i.e., outside of the red region), attain high values as integrals over areas of high magnitude. In contrast, geodesics near the data manifold would low values.

We visualize the decoder-induced geodesic distance and compare it to the latent Euclidean distance in Figures 2b and 2c, respectively. The plots depict the normalized distances of all states to the state marked by a yellow square. Evidently, the geodesic distance captures a better notion of distance in the said environment, correctly exposing the "land distance" in ambient space. As expected, states residing in the bottom-right room are farthest from the source state, as reaching them comprises of passing through at least two bottleneck states. In contrast, the latent Euclidean distance does not properly capture these geodesics, exhibiting a similar distribution of distances in other rooms. Nevertheless, both geodesic and Euclidean distances reveal the intrinsic topological structure of the environment, that of which is not captured by the extrinsic coordinates $(x, y) \in [-1, 1]^2$. Particularly, the state coordinates $(x, y)$ would wrongly assign short distances to states across impassible walls or obstacles, i.e., measuring the "air distance".

**Intersection.** We visualized our metric in the intersection environment proposed in Leurent [2018]. Figure 3 compares the Euclidean and geodesic distances of a partially trained agent. Unlike the previous toy example, to visualize the inherent manifolds we used t-SNE [Van der Maaten and Hinton, 2008] projections computed with Euclidean distance and compared them to the projection computed with geodesic distance, i.e., curves of minimum energy in ambient space (Definition 2). Indeed, the geodesics captured the inherent structure of the environment, whereas Euclidean distances only managed to capture general clusters. These suggest that Euclidean distance might not be representative for measuring distance in latent space, as will also become evident by our experiments in the following subsections.

## 5.2 Datasets and Implementation Details

We used D4RL [Fu et al., 2020] and the autonomous vehicle environments highway-env [Leurent, 2018] as benchmarks for all of our experiments. We tested GELATO on three Mujoco [Todorov et al., 2012] environments (Hopper, Walker2d, Halfcheetah) on datasets generated by a single policy and a mixture of two policies. Specifically, we used datasets generated by a random agent (1M samples), a partially trained agent, i.e, medium agent (1M samples), and a mixture of partially trained and expert

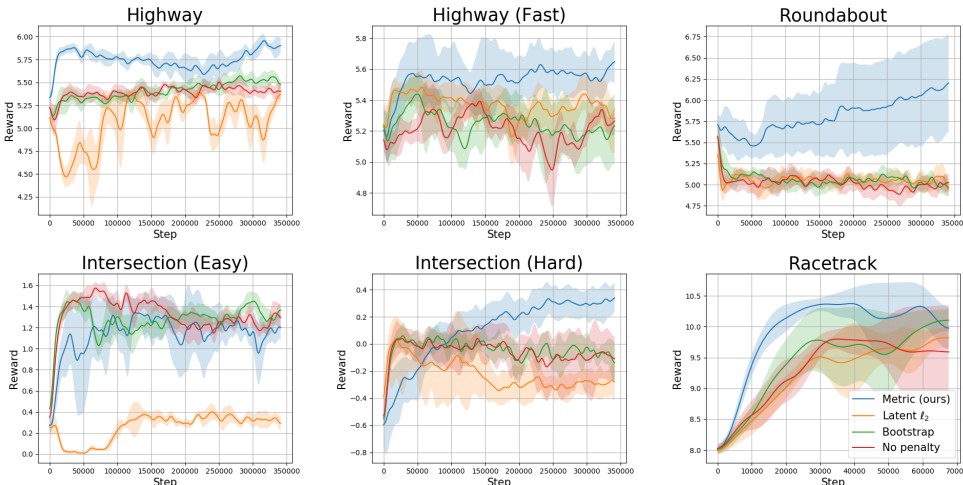

Figure 4: Performance of GELATO with different uncertainty estimation methods on the highway-env benchmarks. Results suggest our induced semiparametric distance is an effective penalty for offline RL.

agents (2M samples). For autonomous driving, we tested GELATO on four environments (Highway, Roundabout, Intersection, Racetrack), on datasets containing five episodes generated by a partially trained agent. We also tested a faster ($\times 15$ speedup) variant of the Highway environment, as well as a harder instantiation of the Intersection environment in which the number of cars was tripled (further details can be found in the appendix).

We trained our variational latent model in two phases. First, the model was fully trained using a calibrated Gaussian decoder [Rybkin et al., 2020]. Specifically, a maximum-likelihood estimate of the variance was used $\sigma^* = \text{MSE}(\mu, \hat{\mu}) \in \arg\max_\sigma \mathcal{N}(\hat{\mu}|\mu, \sigma^2 I)$. Then, in the second stage we fit the variance decoder networks. Hyperparameters for training are found in the appendix.

In order to practically estimate the geodesic distance in Algorithm 1, we defined a parametric curve in latent space and used gradient descent to minimize the curve's energy. The resulting curve and pullback metric were then used to calculate the geodesic distance by a numerical estimate of the curve length (See Appendix for an exhaustive overview of the estimation method).

We used FAISS [Johnson et al., 2019] for efficient GPU-based $k$-nearest neighbors calculation. We set $k = 5$ neighbors for the penalized reward (Equation (3)). Finally, we used a variant of Soft Learning, as proposed by Yu et al. [2020] as our RL algorithm for the continuous control benchmarks, and PPO [Schulman et al., 2017] for the autonomous driving tasks. All agents were trained for 1M steps (for continuous control benchmarks) and 350K steps (for the driving benchmarks), using a single GPU (RTX 2080), and averaged over 5 seeds (see Appendix for more details).

### 5.3 Results

**D4RL.** Results for the continuous control domains are shown in Table 1. We performed experiments to analyze GELATO on various continuous control datasets. We compared GELATO to contemporary model-based offline RL approaches; namely, MOPO [Yu et al., 2020] and MBPO [Janner et al., 2019], as well as the standard baselines of SAC [Haarnoja et al., 2018] and imitation (behavioral cloning, Bain and Sammut [1995], Ross et al. [2011]).

We found performance increase on most domains, and most significantly in the medium domain, i.e., partially trained agent. Since the medium dataset contained average behavior, combining the nonparametric nearest-neighbor uncertainty method with the bootstrap of decoders benefited the agent's overall performance. In addition, GELATO with the latent $\ell_2$ distance metric performed well on many of the benchmarks. We conjecture this is due to the inherent Euclidean nature of the continuous control benchmarks. Unlike the embedding for the autonomous driving benchmarks (Figure 3), we found the D4RL data to project similarly when $\ell_2$ and geodesic distances were used (we provide plots of these embeddings in the Appendix).

**Highway-Env.** Figure 4 shows results for the autonomous driving benchmarks in highway-env. In contrast to the continuous control benchmarks, we found a significant improvement of our metric on the autonomous driving benchmarks compared to standard uncertainty estimation methods as well as the latent Euclidean distance. We credit this improvement to the non-euclidean nature of the environments, as previously described in Figure 3. While Euclidean distances were useful in the Mujoco environments, they performed distinctly worse in the autonomous driving environments.

Our results emphasize the importance of OOD uncertainty estimations methods in reinforcement learning on various types of datasets. While robotic control tasks provided useful insights, they did not capture the non-euclidean nature inherent in alternative tasks, such as autonomous driving.

## 6    Related Work

**Offline Reinforcement Learning.** The field of offline RL has recently received much attention as several algorithmic approaches were able to surpass standard off-policy algorithms. Value-based online algorithms do not perform well under highly off-policy batch data [Fujimoto et al., 2019, Kumar et al., 2019, Fu et al., 2019, Fedus et al., 2020, Agarwal et al., 2020], much due to issues with bootstrapping from out-of-distribution (OOD) samples. These issues become more prominent in the offline setting, as new samples cannot be acquired.

Several works on offline RL have shown improved performance on standard continuous control benchmarks [Laroche et al., 2019, Kumar et al., 2019, Fujimoto et al., 2019, Chen et al., 2020b, Swazinna et al., 2020, Kidambi et al., 2020, Yu et al., 2020]. This work focused on model-based approaches [Yu et al., 2020, Kidambi et al., 2020], in which the agent is incentivized to remain close to areas of low uncertainty. Our work focused on controlling uncertainty estimation in high dimensional environments. Our methodology utilized recent success on the geometry of deep generative models [Arvanitidis et al., 2018, 2020], proposing an alternative approach to uncertainty estimation.

**Representation Learning.** Representation learning seeks to find an appropriate representation of data for performing a machine-learning task [Goodfellow et al., 2016]. Variational Auto Encoders [Kingma and Welling, 2013, Rezende et al., 2014] have been a popular representation learning technique, particularly in unsupervised learning regimes [Chen et al., 2016, Van Den Oord et al., 2017, Hsu et al., 2017, Serban et al., 2017, Engel et al., 2017, Bojanowski et al., 2018, Ding et al., 2020], though also in supervised learning and reinforcement learning [Hausman et al., 2018, Li et al., 2019, Petangoda et al., 2019, Hafner et al., 2019]. Particularly, variational models have been shown able to derive successful behaviors in high dimensional benchmarks [Hafner et al., 2020].

Various representation techniques in reinforcement learning have also proposed to disentangle representation of both states [Engel and Mannor, 2001, Littman and Sutton, 2002, Stooke et al., 2020, Zhu et al., 2020], and actions [Tennenholtz and Mannor, 2019, Chandak et al., 2019]. These allow for the abstraction of states and actions to significantly decrease computation requirements, by e.g., decreasing the effective dimensionality of the action space [Tennenholtz and Mannor, 2019]. Unlike previous work, GELATO is focused on a semiparametric approach for uncertainty estimation, enhancing offline reinforcement learning performance.

## 7    Discussion and Future Work

This work presented a metric for model dynamics and its application to offline reinforcement learning. While our metric showed supportive evidence of improvement in model-based offline RL we note that it was significantly slower – comparably, 5 times slower than using the decoder's variance for uncertainty estimation. The apparent slowdown in performance was mostly due to computation of the geodesic distance. Improvement in this area may utilize techniques for efficient geodesic estimation.

We conclude by noting possible future applications of our work. In Section 5.1 we demonstrated the inherent geometry our model had captured, its corresponding metric, and geodesics. Still, in this work we focused specifically on metrics related to the decoded state. In fact, a similar derivation to Theorem 2 could be applied to other modeled statistics, e.g., Q-values, rewards, future actions, and more. Each distinct statistic would induce its own unique metric w.r.t. its respective probability measure. Particularly, this concept may benefit a vast array of applications in continuous or large state and action spaces.

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
