# OpenReview forum: "Uncertainty Estimation Using Riemannian Model Dynamics for Offline Reinforcement Learning"
_NeurIPS.cc/2022/Conference — NeurIPS 2022 Accept_

### Official Review · Reviewer_TKTA · 2022-07-11

**Rating:** 6
**Confidence:** 4
**Soundness:** 4 excellent
**Presentation:** 3 good
**Contribution:** 3 good

**Summary:**

Using Riemannian distance metrics instead of Euclidean metrics, the authors better estimate uncertainties for out-of-distribution (OOD) data and apply it to model-based Reinforcement Learning (RL) settings. The paper introduces a new pullback metric for such dynamic models, and verifies its performance on several RL benchmarks.

**Questions:**

1) In the pullback metric for model dynamics, how did the authors come up with this choice for metric? How is the time derivative computed? And what about the geodesic computation, are using gradient descent to solve an ODE?

2) In the abstract, the authors mention how the "proposed metric measures both the quality of OOD samples as well as the discrepancy of examples in the data" (line 7). Don't they both in the end mean the same thing?

3) In Figure 1, "high model error" is mentioned twice, but never do we see "low model error". This would be on the manifold, correct? Would it make sense to add it somewhere in the figure/caption?

4) Line 121 mentions how the encoder and decoders are parametric functions, is there any details on this? If normal neural networks are used, would that not be nonparametric?

5) The equation in Definition 2 defines the curve length form Eq. 1, correct? It could be helpful to link these two for better understanding if that's the case.

6) The authors introduce an ensemble of decoders in Line 161, which I assume is then used for all experiments that follow? How about the forward model, do we continue with the assumption in Corollary 1 and practically neglect it? What is the "meaning" of this ambient space X? This could be made clearer. The need for this forward model in the first place is unclear, perhaps Corollary 1 could be entirely removed.

7) The results are compared with a classical bootstrap ensemble method, but isn't that also used in the proposed method for the epistemic uncertainty? What is the difference there with the baseline?

8) Is the four room experiment discrete or continuous? From the actions, it seems to be discrete, but since the latent space is continuous, I was wondering what happens if we sample more points in the latent space, and where they would correspond to in the rooms, i.e. how likely it is they will end up in the walls/obstacles. Also, would the input space be 3D? Isn't a 2D latent space a bit of an easy mapping in this case?

9) Is "Data Score" in Table 1 the score that the collected data achieved? A quick explanation here would help.

10) Why is t-SNE used for the intersection example latent space visualization? Is it because it is more than 2D? How many dimensions is the latent space here?



**Limitations:**

The limitations are properly addressed by the authors.

**Strengths And Weaknesses:**

A very well written paper, where the novelty is clear. The results are quite nice, though showing the improvement in more scenarios would be great, since the authors noticed that the Riemannian manifold metric work better in some specific cases. However, it seems a bit limiting to define the metric for only dynamic systems. The authors could elaborate more on their choice for this specific setting for model-based RL uncertainty estimation as well, as opposed to general uncertainty measurement.

While the whole paper is focused around offline RL, it would be great if online RL can be added into the introduction a bit more so the contrast between the two is clearer. It also isn't clear to the reader why parametric and nonparametric methods are combined for uncertainty estimation, this could be motivated better.

Line 27 the "metric" used for kNN could be associated with how the distance is defined. That way the reader can understand why a "proper metric" is important. Similarly, line 29 could have just a quick explanation on what epistemic and aleatoric uncertainties are. Just model/data uncertainty would help the understanding for the reader.

Section 2.1 is mainly about (online) RL, not sure if the title necessarily fits. Or the third paragraph on offline setting can be expanded and explained more in detail, since it's the focus of this paper. The equation in line 78-79 (which seems quite important) could be elaborated more, in what space the curve length is defined, and why we care about using a curve length for $f(\gamma)$ instead of $\gamma$. Also, it seems section 3 would fit into section 2, since it is also a preliminary/background. Is there a reason it is separate?

Though the Algorithm 1 is presented quite nicely, it could be even further improved if it were a visual figure, I believe. It would make a very strong titlepage figure as well, outlining the novel framework the authors introduce in this paper.

A major issue that can be easily fixed, is that many of the figures, tables, and algorithms are located pages before they are actually referenced in the text.

The whole idea of the metric capturing inherent characteristics of the model dynamics is still a bit vague to me. If it is, for instance, the discrete bottlenecks present in the four rooms example, more samples would be needed to see this clearly. If it is the clustering in the intersection example, more explanation is needed why the clusters for Euclidean distance is not enough. t-SNE visualization is only qualitative, and distances do not necessarily mean much, though maybe some conclusion can be drawn from the shape of the clusters.

A lot of explanation on how the geodesics are found is missing, how the parametric curve is defined for instance. This could be made a much more important part of the paper. The computation time is also mentioned at some point, and this could be described more in detail, exact computation times would be appreciated.

The related works are divided quite nicely, but the last paragraph on disentangling latent space for RL feels a bit out of place and irrelevant.

Lastly, a lot of the discussion was in the results 5.3 already, so the section 7 title could just be conclusion.

All-in-all a well-written paper with plenty of preliminary information for readers of all backgrounds. The proposed framework could be shown using a titlepage figure describing the pipeline/block diagram for more impact. The overall contributions are clear, and the results agree with the conclusions from the authors. Although many of the core ideas follow from previous work, this work enables the application of geometry-aware learning methods in RL context, providing consistently better results than baselines.


Small details:
- There is no appendix, even though it was referenced many times in the paper.
- Missing figure reference in line 19
- Typo line 29 "Riemmannian"
- Line 231 "would low values." misses some word

---

> ### Author Response · Authors · 2022-08-02
> **Response**
>
> Thank you for your thorough and positive review and for your helpful comments and suggestions. We were encouraged that you found our paper to be very well written, our results novel, and our contributions clear. Please find a response to all of your comments and questions below.
>
> Regarding your suggestions on clarifications, we will add these to our paper. Specifically, your comment on lines 27 and 29, Section 2.1, the equation in line 78, and your comments on the related work and discussion.
>
> Re placement of tables and figures: Thank you for your comment. We will rearrange these to make them referenced before they show in the paper.
>
> Re title figure: This is a great idea. We agree with you that an additional figure explaining the algorithm could really benefit clarity of our proposed approach. We will add one to the paper.
>
> Re online / offline: As you suggested, we will add further discussion about the difference between online and offline, and how uncertainty is used in both settings. Particularly, we note that in online RL uncertainty is generally used for optimism, while in offline it is generally used for pessimism, to avoid overestimation of the value.
>
> Re metric capturing inherent characteristics: We understand that this sentence can be a bit vague, and we’ll therefore clarify what we mean in our paper. Specifically, in the four rooms example, we see that the walls act as a barrier in latent space, which is captured by our metric. Regarding the autonomous driving t-SNE plots, we agree that these are only qualitative, and as you mentioned, we only consider their shape to suggest a distinction between our metric and the Euclidean distance.
>
> Re how the geodesic curves are found: Much of this information was pushed to the appendix, but we agree that it is important for it to be explained in the front matter. We will add this information to the paper to make it more clear. Briefly, we defined a parametric curve in latent space and used gradient descent to minimize the curve’s energy.
>
> Answers to Questions:
>
> 1. As we pointed out earlier, we will move information from the appendix to the paper on how the geodesics are computed. We do a numerical estimate of the geodesic, and describe it in Algorithm 2 in the appendix (see supplementary material). We agree this should be explained better in the paper and will add further explanation for clarity. Regarding the metric, it has been used in previous work on generative latent models (see [1], [2]). Intuitively, this metric attempts to find curves of minimum energy in latent space. Since our metric is induced by the ensemble uncertainty (see Theorems 1 and 2), it will search for curves that don’t fall in areas of high uncertainty.
>
> 2. If we understand your question correctly, what we mean is that our metric can capture the discrepancy between examples, while at the same time measure the amount of uncertainty in OOD areas. The discrepancy between examples can also occur for data that is in-distribution.
>
> 3. Yes you are correct, and thank you for your suggestion. We will add a label to make this clear.
>
> 4. This is an important comment that we will clarify in the paper. The models themselves are neural networks, which are parametric. The nonparametric uncertainty comes from using the nearest neighbor method. This makes the method semiparametric (line 327). We will emphasize this point in the paper.
>
> 5. Yes you are correct. We will add a reference to the equation as you suggested.
>
> 6. In practice we use the results of Theorem 2 for uncertainty. Corollary 1 was placed to show a clear decomposition, similar to that of Theorem 1. We will clarify this in the paper.
> Re ambient space - this is the observation space, i.e., state space. We will clarify this as well.
>
> 7. Difference with only using an ensemble: Unlike the pure ensemble method, which uses the variation of the ensemble in observation space, we use the Riemannian metric, induced by the ensemble, to calculate an uncertainty measure based on Equation 3. This is also what makes this metric semiparametric.
>
> 8. Re four rooms example: In this example we used a 2D latent space to help visualize what the latent space is learning, without the need of dimensionality reduction. In general, it would be best to choose a higher dimension. Notice that in the figure we show a mapping of all the points in the environment to their respective latent codes. We believe that with a “denser” environment (i.e., with more states), we would achieve the same structure, though we would need more data.
>
> 9. Yes, the data score is the mean score achieved by the policy in the data. We will add an explanation to the paper.
>
> 10. Yes, we used t-SNE to make visualization easier. This is a qualitative figure, to give a sense of how the latent space looks like. The actual dimension is 32. We will clarify this in the paper.

---

> > ### Author Response · Authors · 2022-08-02
> > **References**
> >
> > References: \
> > [1] Nutan Chen, Alexej Klushyn, Richard Kurle, Xueyan Jiang, Justin Bayer, and Patrick Smagt. Metrics for deep generative models. In International Conference on Artificial Intelligence and Statistics, pages 1540–1550. PMLR, 2018. \
> > [2] Georgios Arvanitidis, Lars Kai Hansen, and Søren Hauberg. Latent space oddity: On the curvature of deep generative models. In 6th International Conference on Learning Representations, ICLR 2018, 2018.

---

> > > ### Comment · Reviewer_TKTA · 2022-08-09
> > > **Thank you for your response**
> > >
> > > Thank you for your time and effort in revising the paper. Regarding the geodesic solve: My apologies for not having seen the supplementary material, yes it is clearly explained there, thank you for the detailed description. I'm not quite certain if Corollary 1 is of enough importance in the main paper, but thank you for addressing all my comments, it helped understanding the work better.

---

### Official Review · Reviewer_Jba5 · 2022-07-12

**Rating:** 6
**Confidence:** 3
**Soundness:** 3 good
**Presentation:** 3 good
**Contribution:** 3 good

**Summary:**

To address the uncertainty quantification (UQ) problem in offline model-based RL, the authors propose a Riemannian metric that captures both uncertainty in dynamics (aleatoric) and OOD data (epistemic). Their offline RL algorithm GELATO (cute name) learns an uncertainty-penalized reward function by learning the geodesic distance through a 'pullback' metric. The geodesic distance is combined with KNN evaluation and bootstrapping to generate improved UQ performance. In control and autonomous driving datasets GELATO achieves good performance.

**Questions:**

Also minor: I found section 4.3 to be a little bit hard to follow at first. $M$ is used to refer to the manifold in the paper except in this section.

**Strengths And Weaknesses:**

Strength:

- Delightfully well written paper. The exposition of motivation, theory, and experiment results are clear and convincing.
- Introduces a new representation for estimating distance of model dynamics in latent space, that is more geometrically salient then Euclidean distance.
- Detailed analysis of success and failure cases in multiple domains in the experiment section.
- Broad applicability of the contribution. As the authors mentioned, their method of capturing geodesics can be applied to many RL estimates such as Q values, rewards, etc. The authors provide a clear and effective way of doing so.

minor details:
- missing reference in line 19, typo of "Riemannian" in line 29.

---

> ### Author Response · Authors · 2022-08-02
> **Response**
>
> Thank you for your positive review! Your comments on the quality of our writing, the contribution of our work, and the significance of the results were encouraging. Also, thank you for pointing out the reference and the typo. Regarding your comment on Section 4.3, we will go over this section a few more times to make sure it is clear.

---

### Official Review · Reviewer_XJeV · 2022-07-20

**Rating:** 7
**Confidence:** 3
**Soundness:** 4 excellent
**Presentation:** 4 excellent
**Contribution:** 4 excellent

**Summary:**

For reinforcement learning agents learned in an offline setting, it is especially important to be able to quantify uncertainty, as the agent may encounter out-of-distribution experiences when deployed in the online environment.  Learning to quantify uncertainty then enables the agent to avoid these regions, by operating under a penalized MDP.  In this work, uncertainty is estimated as a k-nearest neighbor between the current state action pair against the state actions pairs in the offline dataset, according to a distance metric.  The distance metric is formulated by the authors as the geodesic in a learned latent space, modeled as a VAE.  The latent encoder handles aleatoric uncertainty, whereas an ensemble of decoder functions is used to handle epistemic uncertainty.


**Questions:**

I am wondering how flexible the proposed method is to changes in the encoder and decoder beyond the standard Gaussian formulation.  Would the derived pullback metric work for more complex VAEs?  Could similar pullback metrics be derived for more general generative modeling techniques (my understanding of the work of Arvanitidis et al., 2018 is that it is a study on the geometry of generative models in general, not limited to VAEs)?


**Limitations:**

(Social Impact Limitations N/A)

**Strengths And Weaknesses:**

To my knowledge, the paper is quite original.  It builds off of theoretical findings from past work on deep generative models and applies it well to the offline RL domain.

The quality of the paper is high - the idea is straightforward, expressed clearly, and demonstrated cleanly.  The clarity of the writing is excellent, and it was easy to follow what the authors proposed.

I believe this paper has high significance - enabling RL agents trained on offline datasets to quantify their uncertainty is critical for ensuring good continued performance when deployed in an online environment.

---

> ### Author Response · Authors · 2022-08-02
> **Response**
>
> Thank you for your positive review! We were encouraged that you found the clarity and writing of our paper to be excellent, and that you found the paper to be of high significance. Regarding your question, we can definitely derive similar results for more complex distributions, but with the caveat that the pullback metric may not decompose into $G_{\mu} + G_{\sigma}$. There is a benefit to this decomposition since we can capture $\mu$ and $\sigma$ separately, as $\sigma$ captures aleatoric uncertainty. Recall that the pullback metric is defined by a stochastic generator function from the latent space to the ambient (observation) space. We can therefore use any model that induces such a generator. We can also enrich this metric by utilizing prior information, as done in [1].
>
> References: \
> [1] Arvanitidis, Georgios, Soren Hauberg, and Bernhard Schölkopf. "Geometrically Enriched Latent Spaces." In International Conference on Artificial Intelligence and Statistics, pp. 631-639. PMLR, 2021.

---

### Author Response · Authors · 2022-08-02
**A general comment to all reviewers**

We thank the reviewers for spending the time and effort to carefully evaluate our work. We were encouraged by the reviewers’ positive reviews, who found our paper to be of “high quality and high significance” [XJev], “delightfully well written” [Jba5], where the “novelty is clear” [TKTA].  Beyond these encouraging descriptions, the reviewers made valuable comments that we answer in the following.

---

### Meta-Review · Area_Chair_8wrr · 2022-08-28

**Recommendation:** Accept
**Confidence:** Less certain

**Metareview:**

Unanimous accept from 3 reviewers.

I'm uncertain about "accept" given reviewer's XJeV and  Jba5 reviews are on the short and vague side. Reviewer vqqM never responded, even though they would have been a great reviewer for this work (reminded them once and they confirmed, but forgot to follow up again). Reviewer TKTA's review was the most useful, a borderline accept. There is reviewer consensus on novelty, in addition to being well written, with convincing results on MuJoCo and a highway environment.

I myself am unfamiliar with Riemannian metrics/manifolds, however after reading up on the subject, I worried this paper might have been too close to "Latent Space Oddity: on the Curvature of Deep Generative Models" to learn (or compute) latent space metrics, however this works differs by (1) using a variational *forwards* model to consider dynamics, (2) using an ensemble of model to consider epistemic uncertainty, and (3) tying both these aleatoric and epistemic forms of uncertainty into an offline-RL setting, where rewards are pessimistically estimated under uncertainty. While it seems a little ad hoc to suggest this particular method _for_ a particular application (offline RL), which confuses the narrative and motivation a bit, it does seem to give better RL performance in these settings than L2 and ensembling/bootstrapping in Figure 4. This is the most borderline paper I've seen as AC this NeurIPS, but if forced to make a decision, I lean accept.

**Award:**

No

---

### Decision · Program_Chairs · 2022-09-14

Accept